# Monocytic HLA-DR Expression in Immune Responses of Acute Pancreatitis and COVID-19

**DOI:** 10.3390/ijms24043246

**Published:** 2023-02-07

**Authors:** Shiyu Liu, Wenjuan Luo, Peter Szatmary, Xiaoying Zhang, Jing-Wen Lin, Lu Chen, Dan Liu, Robert Sutton, Qing Xia, Tao Jin, Tingting Liu, Wei Huang

**Affiliations:** 1West China Centre of Excellence for Pancreatitis, Institute of Integrated Traditional Chinese and Western Medicine, West China-Liverpool Biomedical Research Centre, West China Hospital, Sichuan University, Chengdu 610041, China; 2Liverpool Pancreatitis Research Group, Institute of Systems, Molecular and Integrative Biology, University of Liverpool, Liverpool L69 3BE, UK; 3State Key Laboratory of Biotherapy, West China Hospital, Sichuan University and Collaborative Innovation Center for Biotherapy, Chengdu 610041, China; 4Department of Respiratory and Critical Care Medicine, Clinical Research Center for Respiratory Disease, West China Hospital, Sichuan University, Chengdu 610041, China

**Keywords:** acute pancreatitis, COVID-19, HLA-DR, monocytes, immune response, immunosuppression

## Abstract

Acute pancreatitis is a common gastrointestinal disease with increasing incidence worldwide. COVID-19 is a potentially life-threatening contagious disease spread throughout the world, caused by severe acute respiratory syndrome coronavirus 2. More severe forms of both diseases exhibit commonalities with dysregulated immune responses resulting in amplified inflammation and susceptibility to infection. Human leucocyte antigen (HLA)-DR, expressed on antigen-presenting cells, acts as an indicator of immune function. Research advances have highlighted the predictive values of monocytic HLA-DR (mHLA-DR) expression for disease severity and infectious complications in both acute pancreatitis and COVID-19 patients. While the regulatory mechanism of altered mHLA-DR expression remains unclear, HLA-DR^−/low^ monocytic myeloid-derived suppressor cells are potent drivers of immunosuppression and poor outcomes in these diseases. Future studies with mHLA-DR-guided enrollment or targeted immunotherapy are warranted in more severe cases of patients with acute pancreatitis and COVID-19.

## 1. Introduction

New insights into the mechanisms of pathology can sometimes arise from similarities between fundamentally different diseases. This effect can be most pronounced during the emergence of a new infectious disease, such as the recent COVID-19 pandemic. One such unlikely pairing is acute pancreatitis (AP) and severe acute respiratory syndrome coronavirus 2 (SARS-CoV-2) infection.

AP is a sterile inflammatory disorder of the pancreas with an increasing global incidence [1] affecting around 2.8 million patients annually [2]. The etiology of AP is diverse and includes gallstones, alcohol excess, hypertriglyceridemia, endoscopic retrograde cholangiopancreatography, certain medicines, and other rarer causes [3]. Most cases of AP patients are mild and uneventful given that supportive care is in time and appropriate. However, some are more severe, which involve local complications (acute pancreatic necrosis or fluid collection; moderately severe acute pancreatitis, MSAP) and/or persistent organ failure (SOFA score of respiratory, circulatory, and renal system equal or more than 2 lasting > 48 h; severe acute pancreatitis, SAP) [4]. Feed-forward auto-amplification of the initial cellular injury in SAP [5,6] results in persistent systemic inflammatory response syndrome (SIRS), multiple organ dysfunction syndrome (MODS), infection, and death. Persistent organ failure [7,8,9,10] and infected pancreatic necrosis [11,12], alone or in combination, are key determinants of severity in AP and contribute to an immune anergy, secondary infections, and a mortality of > 30%. Currently, there are no specific therapies effectively targeting the initial cellular injury or determinants that resulting in MODS [13].

COVID-19, on the other hand, is a potentially lethal infectious disease caused by the enveloped, positive-strand RNA, SARS-CoV-2, affecting over 600 million cases globally [14]. The disease spectrum of COVID-19 is also highly variable, ranging from asymptomatic (test-positive) disease to critical illness (respiratory failure, septic shock, and/or MODS) [15,16]. SARS-CoV-2 mainly utilizes the angiotensin-converting enzyme 2 (ACE2) as the human host cell entry receptor [17], which is ubiquitously expressed in the nasal epithelium, lung, heart, intestine, and kidney and rarely expressed on immune cells [18]. ACE2 is also expressed on pancreatic ductal cells, acinar cells, and islet cells, making the pancreas vulnerable to viral infection [19]. Serum pancreatic enzymes are elevated in 25% of patients suffering COVID-19, which is linked to worsened clinical outcomes including mechanical ventilation and mortality even in those without AP [20,21,22]. Patients with COVID-19 who developed AP during hospitalization also have a more severe clinical course [23], and indeed SARS-CoV-2 may itself precipitate an episode of AP with marked metabolic derangement even in the absence of local complications or organ failure [24]. More importantly, however, patients with severe/critical COVID-19 appear to be increasingly susceptible to secondary infections [25,26] as a result of immune anergy in a similar manner to SAP.

Dysregulated immune responses in SAP and severe/critical COVID-19 have similar patterns of cytokine release and share many pathways of cellular immunity, especially immunosuppression-related monocyte deactivation in the form of downregulated expression of monocytic human leukocyte antigen-DR (HLA-DR) [27,28]. This review summarizes the role of monocytic HLA-DR (mHLA-DR) expression in the development of immunosuppression and organ failure in both SAP and severe/critical COVID-19. 

## 2. Pathogenesis and Immunopathology in AP and COVID-19

### 2.1. Pathophysiological Mechanisms in AP and COVID-19

Diverse stimuli evoke inflammatory cascades with apparently analogous patterns and clinical manifestations, implying similarities in the pathogenesis and symptomatology of AP and COVID-19 [29]. Cytokines and damage-associated molecular patterns (DAMPs), such as histones, high-mobility group box-1 protein, hyaluronan fragments, mitochondrial DNA, and heat-shock proteins are released from dying or injured cells in the injured pancreas or SARS-CoV-2 infected tissues—particularly lungs. This is associated with and results from a series of molecular events, including premature trypsinogen activation, calcium overload, mitochondria failure, endoplasmic reticulum stress, impaired autophagy, or by SARS-CoV-2 proliferation and release, respectively [6,30,31,32,33]. Interaction of DAMPs with pattern-recognition receptors (PRRs), including Toll-like receptors and NLRP3 inflammasome of the adjacent parenchymal cells or immune cells, promotes the production of various pro-inflammatory cytokines and chemokines [31,34,35,36]. Of note, cell death pathways (e.g., autophagy, NETosis, pyroptosis, apoptosis, necroptosis, and ferroptosis) in surrounding immune cells and stromal cells are activated, fueling the cytokine storm and cultivating a positive cell death-inflammation feedback loop [30,37,38]. In COVID-19, virus particles themselves act as pathogen-associated molecular patterns (PAMPs), which could also be identified by PRRs and activate local inflammation and an innate immune response, evoking the cytokine storm and assembling those induced by DAMPs [29,39]. Activated circulating leukocytes, particularly monocytes, are then recruited to the inflamed pancreas or infected lungs, provoking systemic inflammation and organ failure in AP and COVID-19 alike [29,40,41,42,43]. Moreover, monocytes/macrophages could be infected by SARS-CoV-2, triggering massive inflammatory responses in COVID-19 [44]. 

The involvement of adaptive immunity in AP has been recognized, but its precise role in the sterile inflammatory response seen in AP remains poorly characterized [45]. In contrast, SARS-CoV-2 directly activates specific T cell subsets, initiating an adaptive immune response [46]. Persistent viral stimulation, however, leads to T cell exhaustion, with reduced effector functions and proliferative capacity [47]. This T cell exhaustion phenomenon can also be observed in AP patients [48]. 

Levels of several circulating pro-inflammatory cytokines are dramatically elevated and closely correlate with the development of SAP or severe/critical COVID-19 [49,50,51,52]. Patterns of cytokine alterations in AP and COVID-19 were shown to be remarkably similar in a recent meta-analysis, with tumor necrosis factor-alpha (TNF-α), interleukin (IL)-6, IL-8, and IL-10 concentrations significantly higher in more severe forms than non-severe forms of the two diseases [53]. The crosstalk between excessive inflammatory cytokines, platelet activation, complement activation, and endothelial injury forms a deleterious hyper-inflammatory and hyper-coagulopathy environment which is associated with life-threatening complications (i.e., coagulopathy and vascular immune-thrombosis) of AP and COVID-19 [51,54,55,56,57,58].

Systemic lipotoxicity deserves to be highlighted in this context. In severe/critical COVID-19, lipotoxicity can trigger multiple organ failure and mortality resembling SAP [59]. SARS-CoV-2 can directly infect adipose tissue and promotes the release of several inflammatory cytokines [60]. The pancreas itself is a target of SARS-CoV-2, resulting in the interstitial leakage of pancreatic lipase which induces lipolysis of intrapancreatic adipose tissue and release of excess unsaturated fatty acids (UFAs). These toxic UFAs in turn further directly lead to parenchymal cell injury and provoke the release of pro-inflammatory mediators, driving the cytokine storm and organ failure in SAP and severe/critical COVID-19 [59,61,62]. Lipase inhibitors have been shown to ameliorate lipolysis-induced cytokine storms and mortality [61,62,63,64]. 

In summary, the pathophysiological mechanisms of AP and COVID-19 share many similarities including cell death-inflammation cascade, cytokine storms, enhanced lipolysis, and dysregulated immune responses. These immune responses will be discussed in the next section.

### 2.2. Altered Immune Responses in AP and COVID-19

Immune anergy, evidenced by the failure of delayed hypersensitivity responses, correlates with the development of sepsis and mortality in trauma and surgical patients [65,66,67], as well as in SAP [68]. In the first stage of SAP, an excessive pro-inflammatory burst is rapidly followed by an anti-inflammatory reaction that may result in a generalized inflammatory response in sites remote from the initial pancreatic injury site and gives rise to SIRS [69,70,71]. There is a compensatory response to counteract the overwhelming pro-inflammatory state [72], which may ultimately result in immune suppression [73]. In 1996, Bone termed this immunological phenomenon as “compensatory anti-inflammatory response syndrome” (CARS) [65,66,72].

Unlike SIRS, which is clearly defined by clinical parameters, CARS lacks clinical manifestations and can only be defined molecularly by a combination of immunological alterations. In the landmark paper of Volk’s group in 1997, it was described that many septic patients who died from nosocomial infections had associated downregulation of mHLA-DR [74]. Monocytes from these patients had reduced capacity to act in a pro-inflammatory manner by producing TNF-α following stimulation of lipopolysaccharide (LPS) in vitro, termed “immunoparalysis” [74,75]. Where CARS was once thought to follow sequentially from SIRS, current thinking views CARS responses as concomitant to SIRS; balance in both responses restores homeostasis, but an overshoot of the mechanisms of either SIRS or CARS leads to further injury by excessive inflammation or secondary infection and, ultimately, organ failure and death [67,76,77,78,79,80,81,82,83]. Development of CARS results in lymphocyte apoptosis, T lymphocyte anergy, and deactivation of monocytes resulting in reduced mHLA-DR expression. Furthermore, CARS is associated with elevated levels of circulating IL-10, transforming growth factor-beta (TGF-β) and other anti-inflammatory cytokines, which contribute to the risk of secondary infection.

Immune response to SARS-CoV-2 is characterized by the failure of robust type I and type III interferon response and high expression of pro-inflammatory cytokines and chemokines [17]. Like AP, immune alterations, including severe lymphopenia and functional monocyte deactivation, are indicative of immunosuppression in severe/critical COVID-19 patients [84]. Indeed, monocytes exhibit heterogeneous, dynamic, and severity-dependent alterations of transcription and immune phenotype upon acute pathological insults which appear similar in both SAP and severe/critical COVID-19 patients (Figure 1).

Inflammatory monocytes are enriched in the lungs of severe/critical COVID-19 patients and are also the most altered pancreatic immune cells during progression and recovery of AP [85,86]. Decreased monocytic expression of HLA-DR has a predictive value for the poor prognosis of patients with sepsis [87,88], and the level of mHLA-DR expression may identify patients who are susceptible to the development of infectious complications after trauma [89], major surgery [90], and burns [91]. Here, we review the utility of mHLA-DR in assessing the state of the immune response in AP and COVID-19 and detail-relevant implications for therapy.

## 3. Structure and Expression of mHLA-DR

HLA-DR is a type of major histocompatibility complex (MHC) II molecule [92]. It is a heterodimeric glycoprotein composed of the 33–35 kD heavy/α chain and the 27–29 kD light/β chain, assembling into a structure comprising a peptide binding site on top of two immunoglobulin domains [92]. Encoded by adjacent genes, the β chain is polymorphic around the amino acid residues of the peptide-binding site in contrast to the invariant α chain [93]. 

HLA-DR is mostly expressed on antigen-presenting cells (APCs) such as monocytes, macrophages, dendritic cells, and B cells. The primary function of HLA-DR is to present peptide antigens to the immune system for the purpose of eliciting or suppressing T-(helper)-cell responses, eventually leading to the production of antibodies against the same peptide antigen. HLA-D/DR-controlled antigens play an essential role in the cell-to-cell interactions required to generate an immune response [94,95]. 

The biosynthesis, trafficking, and recycling of HLA-DR are regulated by multiple factors affecting cell surface expression. Consequently, the tightly regulated level of HLA-DR expression on the surface of monocytes is thought to be an indicator for monocyte function and the state of the immune response, with high levels of mHLA-DR associated with enhanced antigen presenting capacity and immune activation, and low levels associated with immune suppression.

### 3.1. Measurement of mHLA-DR

Several reviews [67,96,97] have emphasized the importance of flow cytometry as an indicator of immune function in clinical practice. The unit of measurement of HLA-DR via flow cytometer can be the percentage of HLA-DR positive monocytes (%), the mean fluorescence intensity (MFI), the fluorescence unit relative to the monocyte population (RFU), or antibodies per cell (AB/c). Due to the dynamic nature of HLA-DR expression and recycling, it is critical that measurement of expression is standardized. We support the process published by Docke’s and Monnaret’s groups [98,99,100], which have been widely tested and published and appear to result in a strong correlation between transcription and cell surface expression of mHLA-DR. It should be highlighted that a percentage of HLA-DR^+^ monocytes less than 30% or values of AB/c below 5000 represents immunoparalysis, and values greater than 80% or 15 000 AB/c indicate immunocompetence [99]. The critical features for the sampling and measurement of mHLA-DR from human plasma samples are summarized in Figure 2.

### 3.2. Regulation of mHLA-DR Expression

The transcription of mHLA-DR is complex and heterogeneous, mediated by a series of conserved cis-acting regulatory promoter elements and interacting transcription factors [102]. Among these, class II transactivator (CIITA) is the master regulator of HLA-DR transcription [103]. Polymorphisms of CIITA promoter are associated with decreased mHLA-DR expression in patients with septic shock [104]. Besides biosynthesis, the expression of mHLA-DR can be post-translationally regulated by exocytosis, stability, and recycling. The class II-associated Ii peptide (CLIP), generated from cleavage of CD74 (MHC class II invariant chain, Ii) via members of the cathepsin family, is critical for the transport of HLA-DR to the cell surface [105]. In CD74 knockout mice, MHC II molecules are mainly retained in endoplasmic reticulum with reduced levels on the cell surface [106]. Reducing CLIP generation by blocking cysteine protease activity reduced surface MHC II expression, including HLA-DR to 60% on human monocytes in steady state [107]. HLA-DM, the key accessory molecules in the MHC class II loading compartment, catalyzes the dissociation of CLIP in exchange for more stably binding peptides [108]. MHC II molecules on the cell surface are normal in amounts but mainly loaded with CLIP in HLA-DM-deficient mice [109]. HLA-DR loaded with high-affinity peptides are postulated to be more stable than those with CLIP, indicating the role of HLA-DM in regulating mHLA-DR expression [107]. Of note, surface HLA-DR could be internalized, exchanged from lower affinity peptides into other peptides, and rapidly recycled back to the cell surface [110]. In summary, expression of mHLA-DR is finely regulated by multiple steps, including biosynthesis, peptide-loading via cathepsin-induced CLIP and HLA-DM, vesicular transport to the cell surface, and recycling (Figure 3). 

Multiple pro- and anti-inflammatory cytokines are reported to dynamically control the expression of mHLA-DR [112]. The main mechanisms of cytokines modulating HLA-DR expression are summarized in Table 1. However, the detailed regulatory mechanisms of various cytokines on mHLA-DR expression remain largely unknown.

## 4. The Role of mHLA-DR in AP and COVID-19

Monocytic HLA-DR expression alters dynamically in response to the variation of immune responses in the body during the disease course of AP and COVID-19. Evaluating the dynamic expression of mHLA-DR provides indicative information for diagnosis and prediction of disease severity, infectious complications, and prognosis (Figure 4).

The expression of mHLA-DR on admission was downregulated in AP patients compared to healthy controls; it further decreased on days 1 and 2 with differential degrees depending on severity [123,124,125]. While mHLA-DR expression recovered rapidly at day 3 and became normal after day 7 in less severe patients, it persisted at low levels for 1–2 weeks in more severe cases [124,126]. Indeed, mHLA-DR expression displays an inverse relationship with severity throughout at least the first three weeks of disease [127], with the lowest expression of mHLA-DR in SAP consistently recorded between 48 and 72 h of disease onset [127,128].

Overall, mHLA-DR expression either increases or decreases slightly in mild COVID-19 patients compared with healthy controls [129,130]. However, a marked and persistent decrease in expression is described in severe/critical COVID-19 patients in most studies [129,131,132,133,134,135,136,137,138,139,140]. The immune response to severe COVID-19 can be categorized into three groups according to the kinetics of mHLA-DR expression: (i) hyperactivated monocytes/macrophage phenotype (persistently high mHLA-DR > 30,000 AB/c)—strongly associated with mortality; (ii) prolonged immunodepression (persistently low mHLA-DR < 15,000 AB/c after days 5–7)—strongly correlating with secondary infection; (iii) transient immunodepression (early mHLA-DR < 15,000 AB/c, rising above 15,000 AB/c after 5–7 days)—at risk of secondary infection [141]. Patients with acute respiratory distress syndrome (ARDS) secondary to COVID-19 exhibit either immune dysregulation evidenced by very low mHLA-DR expression (i.e., lower than 5000 AB/c) and depletion of lymphocytes, or macrophage activation syndrome characterized by elevated ferritin, where associated HLA-DR levels might be reduced [142], or comparable to healthy controls [143]. Expression of mHLA-DR may be able to provide some information on disease course and has been observed to normalize upon recovery from critical illness in patients with COVID-19 (from 1–3 days to over 10 days after admission), but continued to fall in a patient who died [136]. Critically ill COVID-19 patients with long hospital stays (>25 days) presented with a more profound reduction in mHLA-DR expression than patients with short hospital stays (<25 days) [140]. Furthermore, convalescent COVID-19 patients exhibit mHLA-DR levels which are higher than those of healthy controls at 6 months, and equal to healthy controls at 9 and 12 months following discharge from the hospital [140,144]. 

The reduction of HLA-DR expression in COVID-19 patients has been reported in both classical monocytes [144,145], as well as intermediate monocytes and/or non-classical monocytes [132,146,147], although usually in one group or the other, depending on the respective study. Classical monocytes are the first peripheral immune cell type to recover HLA-DR positivity during the recovery of critically ill COVID-19 patients [148]. 

### 4.1. Severity Prediction Using mHLA-DR in AP and COVID-19

The predictive values of mHLA-DR for severity and mortality of AP and COVID-19 are summarized in Table 2.

MHLA-DR expression is inversely correlated with surrogate biochemical markers of severity (C-reactive protein [CRP], TNF-α, and IL-6) [127,128,154,155,156,157], clinical scoring systems (Ranson, Acute Physiology and Chronic Health Evaluation II [APACHE II], and MODS criteria) [128,154,157,158], and actual severity of AP [125,127,128,150,151,156,158,159,160], and AP patients with low mHLA-DR expression had approximately 2.7 times longer ICU stays than those with normal expression [159]. HLA-DR expressed on classical monocytes was able to distinguish cases of mild from MSAP/SAP and SAP from MSAP on admission [160]. Indeed, mHLA-DR expression on admission, days 2 and 5 all have been shown to have predictive value for SAP [150] and/or the subsequent development of organ failure(s) [151]. 

The utility of mHLA-DR to predict mortality in AP is more controversial. While several studies have reported differences in mHLA-DR expression between survivors and non-survivors on days 7 [124,128] or 10 after admission [152], others found no difference [127,158,161]. These results might be explained by the differences in the design of the respective clinical studies, or by the heterogeneous and dynamic immune response in the study populations. 

Despite one study finding that mHLA-DR expression was irrelevant to severity of COVID-19 [146], most studies demonstrate an inverse relationship [135,136,137,139]. Low or very low mHLA-DR expression has been described in association with ARDS [162], severe respiratory failure [142], thrombocytopenia, increased antibiotic requirements, and need for extracorporeal membrane oxygenation [134,135,142,153,162]. Similarly to AP, lower levels of mHLA-DR expression correlated with length of hospital stay [140], SOFA score [153], and serum clinical biochemical parameters including D-dimer, lactate dehydrogenase, CRP, procalcitonin, ferritin, IL-6, IL-10, granulocyte colony-stimulating factor, chemokine C-X-C motif ligand 10, chemokine C-C motif ligand 2 (CCL2), and IFN-γ levels [135,136,137,144,147,153,163]. Although overall mHLA-DR expression does not appear to differ between survivors and non-survivors [141], the lowest levels of mHLA-DR expression can be observed in patients with COVID-19 who died in the ICU [84]. The proportion of mHLA-DR^+^ monocytes was also lower in deceased COVID-19 patients compared with time-matched controls [164]. Expression of mHLA-DR recovers with clinical improvement but continues to fall in patients who do not survive [134,144].

### 4.2. Prediction of Infectious Complications using mHLA-DR

MHLA-DR regulates the interplay between innate and adaptive immunity and represents an overview of an organism’s capacity for antigen presentation, cytokine production, and phagocytosis [165]. HLA-DR downregulation is not only limited to the blood compartment but can also be observed in lymphatic tissue [166]. With standardization of flow cytometry-based measurement of mHLA-DR, a multicenter comparison of obtained results becomes feasible [98]. Therefore, mHLA-DR is now the most frequently utilized biomarker for assessing the development of immunosuppression in critically ill patients, including sepsis, stroke, trauma, and burns [167]. Following the rationale of immunosuppression in severe critical illnesses, mHLA-DR expression levels are predictive of septic complications in AP and COVID-19 using values shown in Table 3.

Failure of the intestinal barrier function is often thought to be responsible for the dysregulated systemic inflammation in AP [126,168]. The proportion of HLA-DR^+^ monocytes correlated negatively with measures of small intestinal permeability, including the urinary lactulose/mannitol ratio and D(-)-lactate concentrations [126]. AP patients with infectious complications, including sepsis or infected pancreatic necrosis, had lower HLA-DR expression which recovered at a slower rate than those without [127,128,150,152,154,157,169,170,171]. The relative risk of developing infected pancreatic necrosis in AP patients with low mHLA-DR expression that persisted into the second week of illness was 11.3 (1.6–82.4) [170], and persistently low HLA-DR levels have even been shown to be related to multidrug resistant infection [171]. This ability to identify patients with infectious complications early was as good or superior to routine biochemical markers and clinical scoring systems including CRP and APACHE II [152]. Therefore, a persistently low expression of mHLA-DR might be an effective and reliable indicator of potentially lethal infectious complications in patients with AP that could perhaps be used to identify patients who might benefit from early antimicrobial therapy.

As in AP, persistently low levels of mHLA-DR expression are associated with secondary infection in COVID-19 patients [172]. COVID-19 patients who developed secondary bacterial infections exhibit lower levels of mHLA-DR expression than those without at all time points (days 1, 4, and 7 [173], days 5–7, days 8–10 [141]). MHLA-DR expression (AB/c) on days 0–3 and on days 7–10 have been shown to predict secondary infection in COVID-19 patients in the ICU [134]. 

### 4.3. Regulation of mHLA-DR Expression in AP and COVID-19

Both pro- and anti-inflammatory cytokines, including TNF-α, IL-6, IL-8, IL-10, and IL-1RA1, can downregulate—and correlate inversely with—mHLA-DR expression in AP patients [156,160]. IL-6, IL-8, IL-10, and IL-1RA1 inhibit HLA-DR expression on classical monocytes in vitro [160]. TNF-α enhances IL-10 production of monocytes in vitro and downregulates levels of HLA-DR, even in the presence of anti-IL-10 monoclonal antibodies, demonstrating inhibition of mHLA-DR expression via an alternate pathway [156]. 

The regulatory mechanisms of reduced mHLA-DR expression in severe/critical COVID-19 patients are less well understood, but IL-6 and IL-10 are similarly thought to be possible drivers to reduce mHLA-DR expression in the disease. MHLA-DR expression was strongly reduced by plasma from COVID-19 patients with immune dysregulation but not healthy controls [130,142]; the effect could be partially restored by the addition of the IL-6 blocker Tocilizumab [142]. The highly expressed cytokines in COVID-19 patients included IL-10, IL-6, IL-7, TNF-α, IFN-α, CCL2, and CCL4, but only incubation monocytes with IL-10 downregulated HLA-DR expression [130]. 

The altered cytokine profiles in sterile or infectious inflammatory diseases including AP and COVID-19 are dynamic and complex, which may affect mHLA-DR expression synergistically or antagonistically. Future studies are needed to investigate the precise role of cytokines in regulating mHLA-DR so as to develop potential therapeutic targets in immune regulation. 

### 4.4. Monocytic Myeloid-Derived Suppressor Cells

A proportion of circulating HLA-DR^−/low^ monocytes seen in both AP and COVID-19 patients have been identified as CD14^+^CD11b^+^HLA-DR^−/low^CD15^−^ monocytic myeloid-derived suppressor cells (M-MDSCs); these cell types may cloud earlier studies on the topic, as they could be misidentified as HLA-DR^−/low^ classical monocytes [174,175,176]. M-MDSCs are characterized by their potent immunosuppressive effects on other immune cells, especially T cells, through various mechanisms including secretion of arginase-1 (Arg-1), and inducible nitric oxide synthase, production of reactive oxygen and nitrogen species, secretion of cytokines including TGF-β and IL-10, and induction of regulatory T cells [175]. 

The proportion of M-MDSCs in peripheral blood mononuclear cells correlates with AP severity as reflected by plasma CRP levels, APACHE II score, and length of stay [174]. Increased levels of Arg-1 and ROS can further be observed in AP patients, especially those with a severe clinical course [174]. Similarly, expansion of M-MDSCs was reported together with increased Arg-1 activity in plasma, and these are associated with severity and fatal outcome in COVID-19 patients [175].

Therapeutic approaches aimed at reducing the number, function, and accumulation of M-MDSCs might improve the suppressive state of the immune system and improve complication-free survival in both SAP and severe/critical COVID-19 patients. 

## 5. Conclusions and Future Prospects

MHLA-DR expression serves as a useful biomarker for immune (dys)function in patients with AP and COVID-19. The measurement of patterns and dynamics of mHLA-DR expression in both these diseases can help clinicians to determine the severity and prognosis, and perhaps guide timing and selection of therapy. Monitoring mHLA-DR expression appears to help identify and differentiate patients at higher risk of secondary infections associated with poor outcomes. While immunosuppression in general is thought to represent later stages of both diseases, in actual fact, time course and immune responses can be highly heterogeneous and variable [127,177]. MHLA-DR modulation occurs over several days [178], necessitating multiple, consecutive mHLA-DR measurements following a standardized assessment procedure of flow cytometry in patients from point of admission. MHLA-DR measurement should be prioritized for patients with clinically severe presentations with rapidly worsening organ dysfunctions or who are in need of invasive treatments or are at high risk of infectious complications with poor prognosis [179,180,181].

Examples of potential mHLA-DR-directed interventions that could find utility in AP and COVID-19 include several immunostimulatory agents, including IFN-γ [155,182,183,184], recombinant IL-7 [185], and granulocyte-macrophage colony-stimulating factor [155]. Thymosin alpha 1 (Tα1), a peptide hormone used to stimulate the T-cell mediated immune response, has been tested in patients with predicted for necrotizing pancreatitis (presumably immunocompromised), but results are so far disappointing [186]; thus far, there has been no demonstrable reduction in the incidence of infected pancreatic necrosis, new-onset organ failure, or any other complications. Defining immunosuppression, for example, by using the measurement of mHLA-DR expression to guide participant selection and/or tailor the treatment dose, may be required to demonstrate effective immune-stimulatory therapy.

The complex and highly variable immune alterations seen in severe acute inflammation and infection warrant stratified immunotherapy. MHLA-DR expression provides supportive information in determining the timing and strategies of individual immune treatments, including anti-inflammatory, immune-stimulatory or immune-modulatory agents at different disease stages, something that has been demonstrated in both acute pancreatitis and COVID-19. The emergence of a new global pandemic disease has provided valuable insights into the mechanisms of a long-established illness, with considerable potential to draw insights into one disease from the other. There is a need for a simple, cheap, and effective universal immune assessment tool, combining mHLA-DR with established clinical markers of disease severity and possibly other circulating immune cell profiles to aid assessment of the disease course of illnesses with a systemic inflammatory component in order to predict outcomes and to guide treatment decisions.

## Figures and Tables

**Figure 1 ijms-24-03246-f001:**
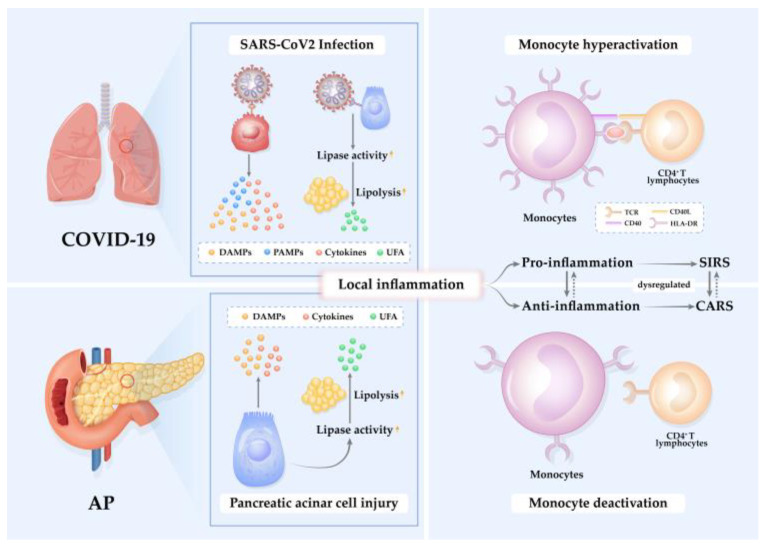
Pathogenesis of inflammation in AP and COVID-19. Acute pathological insults of SARS-CoV-2 infection and pancreatic acinar cell injury elicit local inflammation mediated by cytokines, unsaturated fatty acids (UFAs), damage-associated molecular patterns (DAMPs), and/or pathogen-associated molecular patterns (PAMPs). The pro-inflammatory reaction induces an anti-inflammatory response to restrict inflammation. When the pro-/anti-inflammation is unbalanced and dysregulated, systemic inflammatory response syndrome (SIRS) or compensatory anti-inflammatory response syndrome (CARS) occurs. During SIRS, monocytes are hyperactivated in response to high levels of pro-inflammatory cytokines and chemokines. In contrast, during CARS, monocytes are deactivated, exhibit reduced mHLA-DR expression, and are incapable of presenting antigens to activate CD4^+^ T lymphocytes.

**Figure 2 ijms-24-03246-f002:**
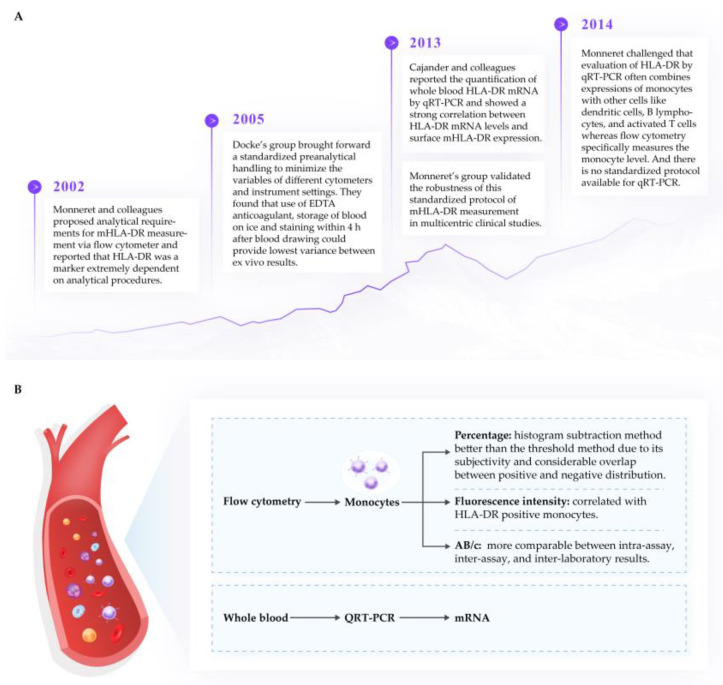
Measurement of mHLA-DR expression. (**A**) Measurement of mHLA-DR expression and requirements for sample handling procedures [97,98,99,100,101]. (**B**) Relationship of units of mHLA-DR expression to different measurement methods.

**Figure 3 ijms-24-03246-f003:**
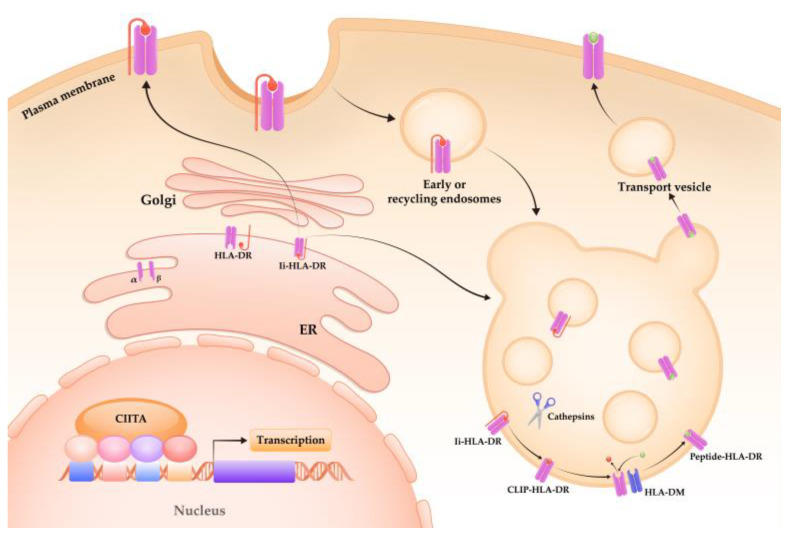
Regulation of mHLA-DR expression. Figure referenced from [105,111]. The transcription of HLA-DR is tightly regulated by a set of cis-acting regulatory promoter elements and transcription factors. Class II transactivator (CIITA) is the master transcriptional regulator. The α- and β-chains of HLA-DR assemble in the endoplasmic reticulum (ER) and then bind with the invariant chain (Ii). The Ii–HLA-DR complexes transport through Golgi complex to the MHC class II compartment (MIIC), directly or via the internalization of the plasma membrane. Ii is degraded into class II-associated Ii peptide (CLIP) via members of cathepsin family. In the aid of chaperone HLA-DM, CLIP is exchanged for antigen peptide. Peptide-HLA-DR complexes are then transported to the plasma membrane for further T cell activation. Interfering with the expression and activity of CIITA, Ii, cathepsins, HLA-DM, as well as the associated vesicle traffic, all result in alteration of the mHLA-DR expression.

**Figure 4 ijms-24-03246-f004:**
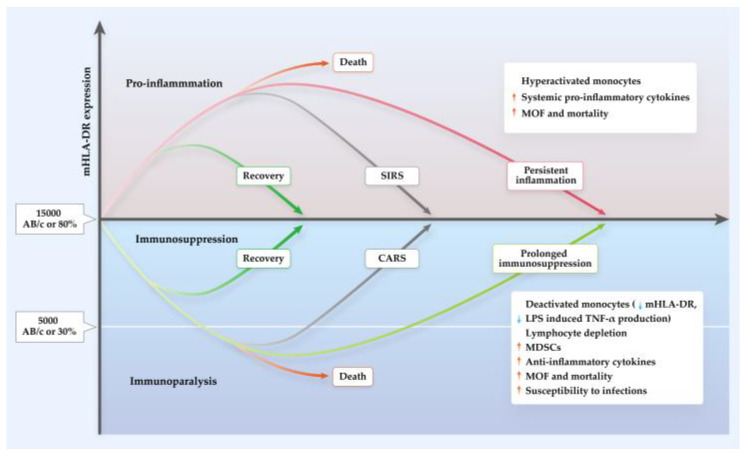
Immune response in AP and COVID-19. Figure referenced from [121,122]. Pro- and anti-inflammatory response are both activated after acute insults of either AP or COVID-19. The initial generalized inflammation is individually heterogeneous. Patients with less intense generalized inflammatory response may survive and restore the immune balance. When inflammation markedly outpaces anti-inflammation, monocytes are hyperactivated, leading to increased systemic release of pro-inflammatory cytokines and resulting in systemic inflammatory response syndrome (SIRS). This cytokine storm and hyperinflammation is associated with multiple organ failure (MOF) and mortality in AP and COVID-19. Conversely, compensatory anti-inflammatory response syndrome (CARS) happens when the anti-inflammatory response is overwhelming. mHLA-DR is an indicator of this, and expression below 15,000 AB/c or 80% characterizes immunosuppression and below 5000 AB/c or 30% characterizes immunoparalysis. In addition to pronouncedly reduced mHLA-DR expression, monocytes are deactivated with TNF-α production upon lipopolysaccharide (LPS) stimulation in CARS. Lymphocytes are depleted, accompanied by a massive release of anti-inflammatory cytokine. This dysregulated and persistent immunosuppression contributes to MOF, death, and infections.

**Table 1 ijms-24-03246-t001:** Cytokine Modulation of HLA-DR Expression.

Cytokines	HLA-DR Expression	Regulatory Mechanisms	References
IL-10	↓	Downregulation of CIITA; Altering vesicular traffic of HLA-DR in exocytosis and recycling	[112,113]
TGF-β	↓	Inhibition of CIITA and downregulation of HLA-DR transcription	[114,115]
IFN-β	↓	Downregulation of CIITA	[116]
IFN-γ	↑	Promotion of HLA-DR and CD74 transcription	[117,118]
GM-CSF	↑	Promoting exocytosis and reducing internalization	[119]
TNF-α, IL-1	↑	Boosting biosynthesis and stability of HLA-DR increasing half-life from about 10 h to over 100 h	[120]
IL-4	↑	Upregulation of CIITA	[113]

Abbreviations: IL, interleukin; CIITA, class II transactivator; TGF-β, transforming growth factor-beta; IFN, interferon; GM-CSF, granulocyte-macrophage colony-stimulating factor; TNF-α, tumor necrosis factor-alpha.

**Table 2 ijms-24-03246-t002:** Predictive Values of mHLA-DR for severity and mortality in AP and COVID-19.

Disease	Prediction	Sample Size (Incidents/Total)	Measuring Time	Cut-Off Value	AUC	Sensitivity	Specificity	Others
AP [149]	MAP from MSAP/SAP	27/50	Admission	<2274 MFI	0.805	70.4%	82.6%	Combined with classical monocyte proportions (AUC, 0.862)
	SAP from MSAP	9/23		<1094.5 MFI	0.690	85.7%	55.6%	-
AP [150]	SAP	19/58	Admission	<50.8%	0.728	72%	72%	-
			Day 2	<43.35%	0.800	84%	80%	-
			Day 5	<60.8%	0.877	82%	78%	-
AP [151]	Organ failure	29/310	Admission	<78%	0.78	83%	72%	-
				<38 RFU	0.81	69%	84%	-
AP [152]	Mortality	7/25	Day 10	<52.3%	0.944	94.4%	85.7%	-
COVID-19 [136]	Critical COVID-19	9/32	Days 0–3	<81.55%	0.961	100.00%	80.00%	-
COVID-19 [139]	Severe COVID-19	48/97	Admission	<143 MFI	0.9	89.6%	81.6%	Independent predictor of COVID-19 severity (OR = 0.976, 95% CI: 0.955–0.997)
COVID-19 [134]	Mortality	35/124	Days 0–3	<11,312 AB/c	0.64	74%	54%	-
			Days 7–10	<4672 AB/c	0.85	75%	86%	-
COVID-19 [153]	Mortality	1/12	Admission	<270.56 cells/mL	0.875	100.0%	87.5%	-

Abbreviations: AUC, area under the receiver operating characteristic curve; MAP, mild acute pancreatitis; (M)SAP, (moderately) severe acute pancreatitis; MFI, mean fluorescence intensity; RFU, relative fluorescence unit; OR, odds ratio; CI, confidence interval.

**Table 3 ijms-24-03246-t003:** Predictive values of mHLA-DR for septic complications in AP and COVID-19.

Disease	Prediction	Sample Size (Incidents/Total)	Measuring Time	Cut-Off Value	AUC	Sensitivity	Specificity	Others
AP [154]	Sepsis	6/64	Admission	<60%	-	100%	91.3%	Superior to Ranson’s score and APACHE II score
			Day 7		-	100%	93.2%	-
			Day 14		-	100%	98.2%	-
AP [128]	Septic complications	11/74	Day 7	<40%	-	73%	94%	-
			Day 10		-	82%	98%	-
			Day 14		-	100%	100%	-
AP [152]	Septic complications	6/25	Day 10	<58%	0.926	76.5%	100%	Comparable to Ranson and APACHE II scores and better than CRP levels (AUC, 0.841, 0.869, and 0.460, respectively)
AP [157]	Secondary infection	11/40	Admission	<35.8%	0.837	81.8%	82.8%	-
COVID-19 [134]	Secondary infection	38/124	Days 0–3	<10,523 AB/c	0.70	76%	60%	-
			Days 7–10	<6804 AB/c	0.62	77%	52%	-

Abbreviations: AUC, area under the receiver operating characteristic curve; APACHE, Acute Physiology and Chronic Health Evaluation; CRP, C-reactive protein.

## Data Availability

Not applicable.

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
