# Peer review of "Monocytic HLA-DR Expression in Immune Responses of Acute Pancreatitis and COVID-19"

_ijms, 2023, doi:10.3390/ijms24043246_

Round 1

Reviewer 1 Report

This is a complete review with an interesting approach regarding the comparison between the pathophysiology of PA and COVID-19. As a result of the similarities, the authors correctly discuss a common therapeutic strategy. The review perfectly covers the literature related to the topic, the figures are of high quality, and the complex information is summarized in tables.

I only recommend including a section comparing in more detail (and extending the information highlighted in Figure 1) basic pathophysiological and mechanistic aspects common in PA and COVID19 (cell death pathways, gene transcription programs involved, regulation of redox state , inflammatory and immune cascade...)

Author Response

Dear Reviewer,

We really appreciate your help with our manuscript. Thank you very much for your professional comments and giving us a chance for further improvement. We have carefully read the comments and revised our manuscript accordingly. Point-to-point responses are marked in sea blue colour in the main text and these changes are displayed as follows:

This is a complete review with an interesting approach regarding the comparison between the pathophysiology of AP and COVID-19. As a result of the similarities, the authors correctly discuss a common therapeutic strategy. The review perfectly covers the literature related to the topic, the figures are of high quality, and the complex information is summarized in tables.

I only recommend including a section comparing in more detail (and extending the information highlighted in Figure 1) basic pathophysiological and mechanistic aspects common in PA and COVID19 (cell death pathways, gene transcription programs involved, regulation of redox state, inflammatory and immune cascade...)

Our response: Thanks for your suggestions. We have added a section (Section 2.1) to describe the pathophysiological mechanisms in AP and COVID-19. Changes in the manuscript are in Pages 2-3, Lines 77-129:

“2.1 Pathophysiological Mechanisms in AP and COVID-19

Diverse stimuli evoke inflammatory cascades with apparently analogous patterns and clinical manifestations, implying similarities in the pathogenesis and symptomatology of AP and COVID-19 [29]. Cytokines and damage-associated molecular patterns (DAMPs), such as histones, high mobility group box-1 protein, hyaluronan fragments, mitochondrial DNA and heat-shock proteins are released from dying or injured cells in the injured pancreas or SARS-CoV-2 infected tissues - particularly lungs. This is associated with and results from a series of molecular events including premature trypsinogen activation, calcium overload, mitochondria failure, endoplasmic reticulum stress, impaired autophagy, or by SARS-CoV-2 proliferation and release, respectively [6, 30-33]. Interaction of DAMPs with pattern recognition receptors (PRRs) including toll-like receptors and NLRP3 inflammasome of the adjacent parenchymal cells or immune cells promotes the production of various pro-inflammatory cytokines and chemokines [31, 34-36]. Of note, cell death pathways (e.g. autophagy, NETosis, pyroptosis, apoptosis, necroptosis and ferroptosis) in surrounding immune cells and stromal cells are activated, fueling the cytokine storm and cultivating a positive cell death-inflammation feedback loop [30, 37, 38]. In COVID-19, virus particles themselves act as pathogen-associated molecular patterns (PAMPs), which could also be identified by PRRs and activate local inflammation and innate immune response, evoking cytokine storm assembling those induced by DAMPs [29, 39]. Activated circulating leukocytes, particularly monocytes, are then recruited to the inflamed pancreas or infected lungs, provoking systemic inflammation and organ failure in AP and COVID-19 alike [29, 40-43]. Moreover, monocytes/macrophages could be infected by SARS-CoV-2, triggering massive inflammatory responses in COVID-19 [44].

The involvement of adaptive immunity in AP has been recognized, but its precise role in the sterile inflammatory response seen in AP remains poorly characterized [45]. In contrast, SARS-CoV-2 directly activates specific T cell subsets initiating adaptive immune response [46]. Persistent viral stimulation, however, leads to T cell exhaustion with reduced effector functions and proliferative capacity [47]. This T cell exhaustion phenomenon can also be observed in AP patients [48].

Levels of several circulating pro-inflammatory cytokines are dramatically elevated and closely correlate with the development of SAP or severe/critical COVID-19 [49-52]. Patterns of cytokine alterations in AP and COVID-19 were shown to be remarkably similar in a recent meta-analysis, with tumor necrosis factor-alpha (TNF-α), interleukin (IL)-6, IL-8, and IL-10 concentrations significantly higher in more severe forms than non-severe forms of the two diseases [53]. The crosstalk between excessive inflammatory cytokines, platelet activation, complement activation and endothelial injury forms a deleterious hyper-inflammatory and hyper-coagulopathy environment which is associated with life-threatening complications (i.e., coagulopathy and vascular immune-thrombosis) of AP and COVID-19 [51, 54-58].

Systemic lipotoxicity deserves to be highlighted in this context. In severe/critical COVID-19 lipotoxicity can trigger multiple organ failure and mortality resembling SAP [59]. SARS-CoV-2 can directly infect adipose tissue and promotes the release of several inflammatory cytokines [60]. The pancreas itself is a target of SARS-CoV-2, resulting in the interstitial leakage of pancreatic lipase which induces lipolysis of intrapancreatic adipose tissue and release of excess unsaturated fatty acids (UFAs). These toxic UFAs in turn further directly lead to parenchymal cell injury and provoke the release of pro-inflammatory mediators, driving the cytokine storm and organ failure in SAP and severe/critical COVID-19 [59, 61, 62]. Lipase inhibitors have been shown to ameliorate lipolysis-induced cytokine storm and mortality [61-64].

In summary, the pathophysiological mechanisms of AP and COVID-19 share many similarities including cell death-inflammation cascade, cytokine storms, enhanced lipolysis, and dysregulated immune responses. These immune responses will be discussed in the next section.”

Reviewer 2 Report

This is an interesting paper that review the role of Human Leucocyte Antigen (HLA)-DR, expressed on antigen presenting cells and that acts as an indicator of immune function in two major pathologies: acute pancreatitis and COVID-19.  

Nevertheless, some remarks must be carried out and some alterations must be performed:

1. Introduction:

- I propose changing the order of the etiologies of acute pancreatitis to “gallstones, alcohol excess, hypertriglyceridemia, …”

4.2. Prediction of Infectious Complications using mHLA-DR

- In the sentence "a persistently low expression of mHLA-DR might be a unique and specific indicator of potentially lethal infectious complications in patient with AP that could perhaps be used to identify patients who might benefit from early antimicrobial therapy", the authors could substantiate and justify why it is a unique and specific marker.

5. Conclusion and Future Prospects

-  In this section the authors should further explore the clinical applicability of mHLA-DR expression. How to do it, what timing to do it, in an isolated way or together with other markers already known, especially in acute pancreatitis. 

Author Response

Dear Reviewer,

We really appreciate your help with our manuscript. Thank you for your professional comments and giving us a chance for further improvement. We have carefully read the comments and revised our manuscript accordingly. Point-to-point responses are marked in sea blue colour in the main text and these changes are displayed as follows:

This is an interesting paper that review the role of Human Leucocyte Antigen (HLA)-DR, expressed on antigen presenting cells and that acts as an indicator of immune function in two major pathologies: acute pancreatitis and COVID-19.  

 Nevertheless, some remarks must be carried out and some alterations must be performed:

  1. Introduction:

- I propose changing the order of the etiologies of acute pancreatitis to “gallstones, alcohol excess, hypertriglyceridemia, …”

Our response: Many thanks for your careful corrections. We agree and have changed the order as suggested (Page 1, Line 40).

4.2. Prediction of Infectious Complications using mHLA-DR

- In the sentence "a persistently low expression of mHLA-DR might be a unique and specific indicator of potentially lethal infectious complications in patient with AP that could perhaps be used to identify patients who might benefit from early antimicrobial therapy", the authors could substantiate and justify why it is a unique and specific marker.

Our response: Thank you for your suggestions. mHLA-DR is a commonly used and well-recognized biomarker for immune function. However, its uniqueness and specificity in immune state cannot be proved in a very scrupulous way if leaving aside its developed standardized analytical procedures, frequently used assessment and consistently reduced expression during immunosuppression. Therefore, we have changed the seemly inappropriate description “unique and specific” into more accurate words “effective and reliable” (Page 12, Line 382). Besides, we have added a few sentences to further describe the relatively unique role of mHLA-DR expression. Changes in the manuscript are in Page 10, Lines 356-365:

“mHLA-DR regulates the interplay between innate and adaptive immunity and represents an overview of an organism’s capacity for antigen presentation, cytokine production, and phagocytosis [168]. HLA-DR downregulation is not only limited to the blood compartment but can also be observed in lymphatic tissue [169]. With standardization of flow cytometry-based measurement of mHLA-DR, multicenter comparison of obtained results becomes feasible [100]. Therefore, mHLA-DR is now the most frequently utilized biomarker for assessing the development of immunosuppression in critically ill patients, including sepsis, stroke, trauma and burns [170]. Following the rationale of immunosuppression in severe critical illnesses, mHLA-DR expression levels are predictive of septic complications in AP and COVID-19 using values shown in Table 3.”

  1. Conclusion and Future Prospects

-  In this section the authors should further explore the clinical applicability of mHLA-DR expression. How to do it, what timing to do it, in an isolated way or together with other markers already known, especially in acute pancreatitis. 

Our response: Thank you very much for your professional suggestion. We have added a few sentences to demonstrate the clinical applicability of mHLA-DR expression as suggested.

Changes in the manuscript are in Page 13, Lines 435-442 and Page 13, Lines 462 respectively:

“While immunosuppression in general is thought to represent later stages of both diseases, in actual fact time course and immune responses can be highly heterogeneous and variable [180, 181]. mHLA-DR modulation occurs over several days [182], necessitating multiple, consecutive mHLA-DR measurements following a standardized assessment procedure of flow cytometry in patients from point of admission. mHLA-DR measurement should be prioritized for patients with clinically severe presentations with rapidly worsening organ dysfunctions or in need of invasive treatments, who are at high risk of infectious complications with poor prognosis [183-185].”

“…and possibly other circulating immune cell profiles…”